# Tailoring Nanoadsorbent Surfaces: Separation of Rare Earths and Late Transition Metals in Recycling of Magnet Materials

**DOI:** 10.3390/nano12060974

**Published:** 2022-03-16

**Authors:** Ani Vardanyan, Anna Guillon, Tetyana Budnyak, Gulaim A. Seisenbaeva

**Affiliations:** 1Department of Molecular Sciences, Swedish University of Agricultural Sciences, P.O. Box 7015, 75007 Uppsala, Sweden; anna.guillon@slu.se; 2Department of Materials Science and Engineering, Division of Nanotechnology and Functional Materials, Uppsala University, P.O. Box 35, 75103 Uppsala, Sweden; tetyana.budnyak@angstrom.uu.se

**Keywords:** recycling, silica nanoadsorbents, adsorption, REE, LTM, functional ligands

## Abstract

Novel silica-based adsorbents were synthesized by grafting the surface of SiO_2_ nanoparticles with amine and sulfur containing functional groups. Produced nanomaterials were characterized by SEM-EDS, AFM, FTIR, TGA and tested for adsorption and separation of Rare Earth Elements (REE) (Nd^3+^ and Sm^3+^) and Late Transition Metals (LTM) (Ni^2+^ and Co^2+^) in single and mixed solutions. The adsorption equilibrium data analyzed and fitted well to Langmuir isotherm model revealing monolayer adsorption process on homogeneously functionalized silica nanoparticles (NPs). All organo-silicas showed high adsorption capacities ranging between 0.5 and 1.8 mmol/g, depending on the function and the target metal ion. Most of these ligands demonstrated higher affinity towards LTM, related to the nature of the functional groups and their arrangement on the surface of nanoadsorbent.

## 1. Introduction

During the past few decades, there has been a continuous increase in the applications of Rare Earth Elements (REE) and their alloys, making them critical elements for development of modern industries [1]. The main target markets using REEs include magnets, metallurgy, catalysts, polishing powders, batteries, mobile phones and other high-technology gadgets. With their growing demand and continuous supply risk, urban mining of REEs from different end-of-life products and industrial waste has gained increasing attention [2]. As such, permanent magnets, which have various applications in the development of new technological devices as well as for green energy production, are one of the most promising secondary sources of REEs that can be recycled and reused [3]. Common Rare Earth-based magnets include Neodymium–iron–boron (FeNdB) and Samarium–cobalt (SmCo) magnets [4]. One of the key challenges in the recycling of magnet materials lies in the need to separate REE from Late Transition Metals (LTM), which form their constituents or major materials of their casings [5]. The most common technology for REE separation is acidic leaching with different leaching agents such as hydrochloric and sulfuric acids [6,7]. Iron is the major component in FeNdB and in the casing of magnets in electronics. It needs to be removed in the step previous to the separation of all other components, for both economic and technical reasons. The well-established approaches for separation of iron from leachates include either precipitation of iron hydroxide during controlled elevation of pH [8] or the precipitation of all other components apart from iron by addition of oxalic acid and organic base [9]. The precipitate can then be calcined and re-dissolved in acid. Subsequent separation of the components can be achieved by solvent extraction, ion exchange, or adsorption [10]. Most of the established industrial methods require, however, repeated steps to obtain the desired purity, thus generating large amounts of hazardous and, in the case of primary ore treatment, even radioactive waste [11,12]. As an alternative to the traditionally used methods, the application of solid-phase extraction (SPE) using nanosized functional adsorbents has been proven to be an effective and more environmentally friendly method for REE recovery [13]. Various types of nanosorbents have been synthesized for SPE recovery of REEs [14,15,16,17,18,19,20,21]. Several reviews have already been devoted to the discussion of the advantages and challenges in the application of functional solid adsorbents for REE separation [22,23]. Recent works have shown that organic–inorganic functionalized silica nanoparticles possess great adsorption capacity and selectivity towards many metal ions, including heavy metals and REEs [24]. Functional groups such as iminodiacetic acid (IDA), diethylenetriaminepentaacetic acid (DTPA), ethylenediaminetetraacetic acid (EDTA) and triethylenetetraminehexaacetic acid (TTHA) were grafted on dense SiO_2_ nanoparticles as well as SiO_2_ core–shell magnetic nanoparticles and tested for different REEs adsorption and separation. High adsorption capacities of up to 300 mg of RE^3+^/g were reached, and distinct selectivity trends towards different REEs depending on the complexonate [25,26,27].

The majority of earlier successfully applied ligands belonged to the classes of either complexons, i.e., amino carboxylic acids [28], or crown ethers/cyclenes. These types of ligands revealed strong affinity to both REE and LTM. In this paper, the new ligands (N-(2-Aminoethyl)-3-aminopropyltrimethoxysilane **L1** and N-(2-Aminoethyl)-3-AminoIsobutylmethyl Dimethoxysilane **L4**—a derivative of ethylene diamine, known for high affinity to Ni^2+^ and Cu^2+^ and, to a lesser extent, for Co^2+^ [29,30,31]; 2-(2-Pyridilethyl) Thiopropyltrimethoxysilane **L2**—a derivative of pyridine with potentially good affinity to LTM and also a sulfur bridge [32,33]; and Triethoxy(3-isothiocyanatopropyl)silane **L3** and Triethoxy(3-thiocyanatopropyl)silane **L5**—derived from isothiocyanate with potential affinity to Ni^2+^ and Co^2+^ [34], were selected and grafted on SiO_2_ nanoparticles to achieve selectivity for separating LTM from REEs in mixed solutions (Figure 1).

Previously, N-(2-Aminoethyl)-3-aminopropyltrimethoxysilane has been used in modification of silicone tie-paints as a curing component, resulting in significantly improved adhesion between the tie-coating and the epoxy primer [35]. In another work by Zhang et al. [36], **L1** was used for lacquer surface modifications, which showed considerably enhanced aging and corrosion resistance properties. To the best of our knowledge, none of these ligands (**L1–5**) have previously been tested for metal uptake and selective adsorption. The functionalized nanoparticles were characterized extensively and their behavior in solution was investigated both for the adsorption and separation of REEs and LTM by selective uptake.

## 2. Materials and Methods

### 2.1. Reagents

For the synthesis of SiO_2_ NPs, Tetraethyl orthosilicate (TEOS) (CAS No. 78-10-4) (99%) and NH_4_OH (25%) were purchased from Sigma Aldrich Sweden (Stockholm, Sweden). Organic ligands N-(2-Aminoethyl)-3-aminopropyltrimethoxysilane (CAS No. 1760-24-3), 2-(2-Pyridilethyl) Thiopropyltrimethoxysilane (CAS No. 29098-72-4), Triethoxy(3-isothiocyanatopropyl)silane (CAS No.58698-89-8), N-(2-Aminoethyl)-3-AminoIsobutylmethyl Dimethoxysilane (CAS No. 23410-40-4) and Triethoxy(3-thiocyanatopropyl)silane (CAS No. 34708-08-2) were purchased from Gelest (Morrisville, PA, USA). Hydrated nitrates of REEs, Neodymium(III) nitrate hexahydrate Nd(NO_3_)_3_·6H_2_O (CAS No. 16454-60-7), Samarium (III) nitrate hexahydrate Sm(NO_3_)_3_·6H_2_O (CAS No. 13759-83-6) and late transition metals Cobalt nitrate(II) hexahydrate Co(NO_3_)_2_·6H_2_O (CAS No. 10026-22-9) and Nickel nitrate (II) hexahydrate Ni(NO_3_)_2_·6H_2_O (CAS No. 13478-00-7) were also purchased from Sigma Aldrich, as well as the organic solvents ethanol and toluene. All chemicals used in the experiments were purchased as analytical grade and used without further purification.

### 2.2. Synthesis of SiO_2_ Nanoparticles

Dense silica nanoparticles were synthesized by Stöber method, applying specific reaction parameters described earlier [26], which lead to nanoparticles with desired size and shape (≈80 nm). Tetraethyl orthosilicate (TEOS) was used as a precursor and hydrolysis was performed in alcoholic solution using ammonium hydroxide as a catalyst. For that, a mixture of 200 mL ethanol, 35 mL Milli-Q water and 7.5 mL of NH_3_ 25% was prepared in a reaction flask and set to 70 °C with a condenser at nitrogen atmosphere. Afterwards, 11.16 mL TEOS was added very slowly (0.25 mL/min) dropwise with the use of a syringe, and the reaction mixture was stirred for 2 h. SiO_2_ nanoparticles were then separated by centrifugation (10,000 rpm, 15 min) and washed 3 times with Milli-Q water and twice with ethanol. 

### 2.3. Surface Functionalization

Surface functionalization with different ligands was performed in a one-step reaction, since all the ligands contained organosilane groups (trimethoxysilane or triethoxysilane). Typically, 500 mg SiO_2_ nanoparticles were dispersed in 20 mL toluene to which 1 mL corresponding ligand was added. The reaction mixture was refluxed for 24 h under inert atmosphere, and afterwards, the nanoparticles were separated by centrifugation (10,000 rpm, 15 min), washed twice with toluene and twice with ethanol, and dried under nitrogen atmosphere. 

To increase the amount of grafted ligands, another series of experiments was performed with the additional step of washing the synthesized nanoparticles with 1 M nitric acid for 1 h before the functionalization step. 

### 2.4. Adsorption Isotherms and Kinetics with Late Transition Metals (LTMs) and REEs

Different concentrations of LTMs and REEs were chosen for isotherm experiment (0.5 mM, 1 mM, 5 mM, 10 mM and 20 mM). 1 M NaNO_3_ was added to the metal solutions to a final concentration of 0.1 M, to keep the ionic strength constant throughout the experiment. The pH of the solutions was set using 0.1 M HNO_3_ or 1.34 M ammonia and measured with pH meter (Accumet AE150, Fisher Scientific, Hampton, NH, USA) before (pH = 5.8) and after the adsorption test, where no noticeable change was recorded (pH shift after adsorption ≈ 0.2 for all the samples). Then, 20 mg nanoparticles were mixed with 20 mL metal solutions and put on a shaker for 24 h. For kinetic experiments 40 mg of NP samples were mixed in 40 mL metal solution and the metal uptake was checked after set intervals of time. NPs were centrifuged (7000× *g*) for 10 min and 1 mL aliquot was separated to determine the metal concentration in the remaining solution. The samples were first diluted 10 times, and titrated afterwards with EDTA and xylenol orange as an indicator. For each sample the titrations were repeated 3 times, and the average value was calculated. All experiments were performed in triplicate. The relation between the amount of adsorbed metal per unit mass of sorbent and equilibrium concentration of that metal ions is described by adsorption isotherms. Two most frequently used equations correspond to Langmuir and Freundlich isotherm models [37,38]. Previously, linear regression has been regarded as a major preference in designing adsorption processes. However, recent studies have shown that error distribution can change depending on the approach of isotherm equation linearization [39]. Consequently, it is necessary to imply both linearized and non-linearized methods to compare and determine the best model to fit the experimental data.

The Langmuir adsorption isotherm suggests that uptake occurs on a homogeneous surface by monolayer sorption without interactions between adsorbed molecules. The non-linear and linear forms of Langmuir equation can be written as:$$\frac{C_{e}}{q_{e}}=\frac{1}{q_{max}K_{L}}+\frac{C_{e}}{q_{max}}\;(\mathrm{linear}\;\mathrm{equation})\;q_{e}=\frac{K_{L}q_{max}C_{e}}{1+K_{L}C_{e}}\;(\mathrm{nonlinear}\;\mathrm{equation})$$
where $C_{e}$ is the equilibrium concentration of metal ions (mg/L), $q_{e}$ is the amount of metal adsorbed per specific amount of adsorbent (mg/g), $q_{max}$ is the maximum adsorption capacity of adsorbent (mg/g), and $K_{L}$ is an equilibrium constant that reflects the affinity between the adsorbent and adsorbate (L/mg). The values of $q_{max}$ and $K_{L}$ were calculated by both linear (from the slope and intercept of the linear plot of *C_e_*/*q_e_* versus *C_e_*) and non-linear (by Originpro 9) methods. 

The Freundlich adsorption isotherm describes reversible and multilayer adsorption on a heterogeneous surface, where the adsorbed amount increases with the concentration according to the following non-linear and linearized equations: $$\ln q_{e}=\ln K_{f}+\frac{1}{n}\ln C_{e}\;(\mathrm{linear}\;\mathrm{equation})\;q_{e}=K_{f}C_{e}^{1/n}\;(\mathrm{nonlinear}\;\mathrm{equation})$$
where $q_{e}$ is the amount of metal ions adsorbed per unit mass of sorbent (mg/g), $k_{f}$ is the Freundlich constant related to the adsorption capacity (mg/g), *C_e_* is the concentration of adsorbate in the solution at equilibrium (mg/L) and n is related to the adsorption intensity of the adsorbent (L/mg). In the linear form of Freundlich isotherm, the slope and the intercept of the plot of ln*q_e_* versus ln*C_e_*, correspond to 1/*n* and *K_f_*, respectively. Same values were also calculated via non-linear fitting analysis. 

### 2.5. Selectivity of Functionalized SiO_2_ Nanoparticles

For the selectivity test, 10 mg grafted SiO_2_ NPs were placed in 50 mL Falcon tubes and 10 mL of equimolar (5 mM each metal) metal mixture (Ni/Nd, Co/Sm and Ni/Co) was added afterwards. The tubes were put on a shaker for 24 h and centrifuged afterwards at 7000× *g* for 15 min to separate the metal solutions. The samples were dried at N_2_ atmosphere, and analyzed by energy-dispersive X-ray spectroscopy (EDS) for metal mapping. 

### 2.6. Characterization

Particles were morphologically characterized by Bruker (Billerica, MA, USA) FastScan Bio Atomic Force Microscope (AFM). Scanning Electron Microscopy with Energy Dispersion Spectroscopy (SEM-EDS) analyses were performed using Hitachi (Tokyo, Japan) Flex-SEM 1000 environmental scanning electron microscope combined with Xstream 2 EDS detector by Oxford instruments (Abingdon, UK). For each sample in EDS analyses at least 5 different areas were studied, and the average value was calculated and given as the relative content of the elements.

UV-Vis spectra of adsorbents before and after uptake of Co^2+^ ions were recorded and compared to absorption spectra of pure Cobalt salt in the wavelength range 400–800 nm using a Multiskan Sky High plate reader (Thermo Fisher Scientific, Waltham, MA, USA).

Specific surface area and pore volume were determined from nitrogen adsorption/desorption isotherms at −196 °C (Micromeritics ASAP 2020 Surface Area and Porosity Analyser, Norcross, GA, USA). The samples were degassed at 120 °C during 6 h before the measurements.

Fourier-transform infrared (FTIR) spectra of the functionalized nanoparticles and bare ligands were recorded as KBr pellets and in a liquid form using demountable cell with KBr glasses on PerkinElmer Spectrum 100 instrument. Thermogravimetric analyses (TGA) were carried out using a PerkinElmer (Waltham, MA, USA) Pyris 1 instrument in an air atmosphere at a heating rate of 5 degrees/min in the 25–900 °C interval.

## 3. Results and Discussion

### 3.1. Characterization

The morphology of the synthesized samples was analyzed by AFM, revealing NP size varying in the range 80–90 nm (Figure 2). It was also established that acid treatment and subsequent functionalization of the NPs did not influence the morphology or the size of the NPs (Appendix A).

Brunauer–Emmett–Teller specific surface area (SBET) was determined from the low-temperature nitrogen adsorption/desorption isotherms and was found to be 53 m^2^/g for the initial SiO_2_ NPs and 74, 60 and 35 m^2^/g for the modified SiO_2_ NPs: **L2**, **L4**, and **L5_acid**, respectively. For all materials, the shape of the isotherms corresponded to characteristic type II (Figure 3), which is common for non-porous or macroporous materials [40]. The hysteresis loop of isotherms belonged to type H3, which refers to non-rigid aggregates of particles (slit-shaped pores) [40]. The increase of surface area on functionalization can supposedly be attributed to partial de-aggregation of particles. The decrease in surface area for the **L5_acid** sample bearing a larger amount of ligands may occur due to blocking of the pores by ligand molecules. 

Thermogravimetric analyses of functionalized nanoparticles were performed in the temperature range between 20–900 °C at 5 °C/min rate. The TGA curves are summarized in Appendix A. Weight losses of between 3 and 9% were observed for all NPs at 20–200 °C due to the dehydration of the NP surface and evaporation of residual organic solvents (toluene). Above 200 °C, around 9–23% weight loss was observed between 200 and 600 °C, for SiO_2_ NPs functionalized with ligands **L1**–**5** (**SiO_2__L1**, **SiO_2__L2**, **SiO_2__L3**, **SiO_2__L4** and **SiO_2__L5**) (Table 1). The losses above 600 °C were related to carbonization of organic residues on SiO_2_ NPs. These results confirmed that the surfaces of the SiO_2_ NPs were successfully functionalized with the ligands, and the grafted amounts varied depending on the functional groups of the ligands. To increase the amount of functional groups on the silica surface, SiO_2_ NPs were pre-treated by washing with nitric acid. It is known that functionalization of different groups takes place at surface hydroxy sites, and thus it is very dependent on the presence of surface silanol groups [41]. To reach maximum silane capacity, several pre-treatment methods have previously been developed, including washing with acids, piranha solution, and UV-ozonolysis [42]. Herein, we selected three ligands (**L3**, **L4**, **L5**) to functionalize acid-treated SiO_2_ NPs, and the TGA analysis was performed, and the results compared to those of the regular SiO_2_-grafted NPs. 

For **L3** and **L5** ligands, the TGA analysis showed double the increase in the amount of grafted ligands after acid wash, while for **L4**, the grafted ligand amount remained unchanged. **L4** as well as **L1** are nucleophiles, and since amines are basic, a self-catalyzed condensation reaction would be expected, hindering dense attachment on the SiO_2_ NP surface. However, in case of **L4**, it is possible that the faster-reacting ligand undergoes homo-condensation instead of condensing with SiO_2_ NPs, resulting in formation of clusters, which would be washed out during the washing step.

The FTIR results showed that most of the characteristic bands corresponding to ligands were also present in the IR spectra of functionalized SiO_2_ NPs, which further confirms the successful grafting of ligands (Figure 4 and Appendix A). Functionalized SiO_2_ NPs also showed characteristic peaks of SiO_2_ around 460 cm^−1^, 800 cm^−1^ and 1090 cm^−1^, corresponding to δ(Si–O–Si), υ(Si–O–Si) and υ_as_(Si–O–Si) vibrations. Bands at 950 cm^−1^ and 1640 cm^−1^ revealed the presence of residual hydroxyl functions υ_as_(Si–OH) and δ(O–H), respectively. A characteristic band for **L1** appeared at 1470 cm^−1^ corresponding to the amine group, while **L3** and **L5** ligands resulted in the appearance of extra bands at 2100 cm^−1^ and 2150 cm^−1^ that indicated the presence of isothiocyanate and thiocyanate groups, respectively. All grafted nanoparticles (**L1**–**5**) showed peaks at 2990 cm^−1^, which were attributed to stretching vibrations of C–H groups from alkyl chains of the ligands (Figure 4 and Appendix A). Some of the bands also disappeared or overlapped with SiO_2_ characteristic bands after functionalization. The bands at 2944 and 2837 cm^−1^, which could be detected in the ligands’ spectra (Appendix A), corresponded to the stretching of C–H from methoxy-groups present in the ligands before the grafting.

### 3.2. Adsorption Equilibrium Isotherms

The effect of the concentrations of LTM (Co and Ni) and of RE^3+^ (Nd and Sm) on adsorption efficiency was investigated in batch studies at room temperature. The results showed that the adsorbed amount of metal ions on grafted SiO_2_ NPs increased with increasing the initial metal concentration and reached to the maximum adsorption capacity at higher concentrations due to saturation of the binding sites (Figure 5). The maximum adsorption capacities are summarized in Table 2. Based on the results, **SiO_2__L1** demonstrated higher adsorption capacities for all LTM and REEs, which is in agreement with TGA results, considering the grafted amount of the ligands per unit mass of SiO_2_ NPs was higher compared to **SiO_2__L2** and **SiO_2__L3**. SiO_2_ with both **L2** and with **L3** had similar maximum adsorption capacities for most of the metal ions, only for Co adsorption did SiO_2_ NPs grafted with **L3** ligands demonstrate a slightly higher adsorption capacity. However, acid-treated SiO_2_ with **L3** (**SiO_2__L3_acid**) and **L5** (**SiO_2__L5_acid**) showed improved adsorption by increasing their maximum capacities by twice for Co, Nd, Sm, and by almost five times in the case of Ni. According to the literature, sulfur- and amine-containing groups possess higher selectivity towards LTM, which can explain the higher adsorption capacities towards Ni and Co for most of these ligands [43,44,45,46,47,48].

Many silica-based adsorbents have previously been tested for REE and LTM removal from aqueous solutions. Table 3 summarizes various results with silica-based and other adsorbent materials for their maximum uptake capacity. It can be noted that **SiO_2__L*n*** nanoparticles present a competitive performance for the binding of selected metals. Some materials have significantly high sorption properties; however, in most cases, the organosilane-functionalized NPs have comparable or higher adsorption capacities. 

It has to be mentioned that selective separation of REE from LTM by functional nanoadsorbents has so far not been addressed in the literature. 

To understand the adsorption mechanism pathways, and characterize the adsorption equilibria, the experimental data were interpreted by linear (Appendix A) and non-linear Langmuir (Figure 5) and Freundlich (Appendix A) sorption isotherms, and the corresponding parameters were summarized in Appendix A. Consistent with the correlation coefficient (*R*^2^) values, the Langmuir model showed better fit both in linear and non-linear forms, suggesting that the adsorption was a monolayer process on the surface of functionalized silica nanoparticles.

Better insights into the molecular mechanisms of adsorption require application of structure-sensitive characterization techniques. In the present work, we recorded calibrated UV-Vis spectra of adsorbents before and after uptake of LTM. The spectra (Appendix A) reveal a distinct shift of the adsorption maximum for the Co^2+^-derived materials, indicating that uptake of this cation presumably proceeds via formation of inner-sphere complexes with surface ligands. An image demonstrating the potential binding geometry is provided in Figure 1. More detailed insight into coordination of adsorbed cations would require more in-depth investigation with techniques such as, for example, X-ray absorption spectroscopy, applying well-defined structural models from X-ray diffraction studies of relevant model compounds, and will be reported separately later.

The adsorption kinetics showed that most of the uptake (60–80%) occurred within the first 1–2 h of interaction of metals with silica nanoparticles. Slower adsorption continues, and equilibrium is reached after 3–5 h (Figure 6). 

### 3.3. Selectivity and Desorption Tests with Functionalized SiO_2_ Nanoparticles 

EDS spectroscopy was used to analyze the composition of SiO_2_ NPs with different functional groups after adsorption tests with mixtures of metal ions (Nd/Ni, Co/Sm and Ni/Co). Although this technique is able to provide only surface and local information for dense samples, for nano powders such as those used in this work, EDS with its penetration depth of about 1 μm and spot size not less than 100 nm actually provides averaged information about the volume of the material. For most of the samples, we could clearly observe selectivity towards a specific metal ion (Table 4). Thus, for example, SiO_2_ with ligand **L1** (**SiO_2__L1**) demonstrated high selectivity towards Ni ions both in the mixture with REEs and with Co ions. On the other hand, both **SiO_2__L2** and **SiO_2__L3** had better selectivity towards Co ions. The results from EDS analysis with atomic ratios and mapping spectra are summarized in the Appendix A. These results were actually quite unexpected and exciting. The provided functions were supposed to reveal enhanced affinity to LTM, but the experimental results confirmed this supposition only for **L1**, and not **L2**–**5**. The factor that might have been underestimated in building the adsorbent surface is the potential distance between the functions, defined by the grafting density on the silica surface. As earlier studies of molecular models have indicated, the distance between the connection sites for ligand fixation is commonly in the range 5.3–6.6 Å [48], limiting the availability of adsorption sites for especially smaller cations. Higher ligand grafting of **L1** under the applied reaction conditions favored coordination of Ni^2+^ not only via the ethylene diamine site, but probably also via higher ligand density. The lower ligand density for other ligands may have contributed to better adsorption of REE. This was confirmed by grafting the ligands after acid pre-treatment of SiO_2_ NPs, which led to denser functionalized NPs in case of **L3** and **L5**. Subsequent adsorption and selectivity experiments showed that the affinity of these ligands shifted towards LTM (Table 2 and Table 4, Appendix A) by achieving higher maximum capacities and molar ratios for Co (in the case of **SiO_2__L3**) and Ni (in the case of **SiO_2__L5**). 

Desorption tests were performed with 1 M nitric acid in 50 mL Falcon tubes. After adsorption experiments, the samples were centrifuged and 20 mL nitric acid was then added. The tubes were placed on a shaker for 24 h. Afterwards, the nanoparticles were separated by centrifugation (7000× *g*) and the supernatant was collected, neutralized with ammonia, and titrated with EDTA to determine the desorbed metal amount. The results showed that good desorption rates (85–90%) were achieved with all metal ions. Reusability of these NPs was also tested after three cycles of adsorption and desorption of selected metals (Ni and Sm). Desorption was performed using 1 M nitric acid in 50 mL Falcon tubes, mixing the samples for 24 h on a shaker. Afterwards, the samples were centrifuged (7000× *g*), and the supernatant collected, neutralized with ammonia to pH = 6.5, and titrated with EDTA to calculate the amount of desorbed metal. After desorption, the samples were placed in 10 mL metal solutions (Co and Sm respectively) with a final metal concentration of 20 mM. All the samples showed good desorption rates in each cycle ranging between 70 and 100%. A small decrease in adsorption rates was noticed after the second desorption cycle, especially for **L2** and **L4**; however **L1**, **L3** and **L5** had good adsorption/desorption rates even after three cycles for both metals (Table 5 and Table 6).

## 4. Conclusions

New hybrid SiO_2_ NPs were synthesized by one-step grafting, characterized and tested for their adsorption and separation properties in single and mixed metal solutions. As shown by TGA and FTIR analyses, the ligands were successfully grafted on the dense silica NPs. All sorbents showed high adsorption capacities ranging from 0.5 to 1.8 mmol/g for different metal ions. Under dynamic conditions, silica adsorbents revealed fast adsorption kinetics reaching equilibrium after 3 h contact time with metal ions. The selectivity tests demonstrated that the affinity of most of the selected ligands (Ligand 2–5) was towards REE, which was surprising since the functional groups were expected to have higher selectivity towards LTM. However, the second part of adsorption tests with denser functionalized NPs showed a shift towards LTM. This confirmed that higher ligand grafting favored coordination of LTM not only by the functional groups, but also by the distance between the functions. Overall, this work suggests that the evaluated sorbents have the potential for REE separation from LTM as an environmentally friendly alternative to the conventional separation techniques.

## Data Availability

Not applicable.

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
