# Peer review of "Tailoring Nanoadsorbent Surfaces: Separation of Rare Earths and Late Transition Metals in Recycling of Magnet Materials"

_nanomaterials, 2022, doi:10.3390/nano12060974_

Round 1

Reviewer 1 Report

This manuscript reports Tailoring nano adsorbent surface for recycling of Rare Earth-based magnets, though the material synthesis lacks novelty, the results are significant in the field of rare earth element capture. In addition, the prepared materials were characterized satisfactorily. Therefore, this manuscript can be accepted subject to answering the following comments.

  1. Authors are suggested to check the technical and grammatical error in the abstract line 14, adsorption raging 0,5-1,8mmol/g, and all over the manuscript.
  2. How authors were set the pH of adsorption and desorption, details may be needed to understand the process behind the experiment
  3. A Schematic diagram for metal ion binding with ligands can give more understanding to the readers
  4. Authors are suggested to provide the details on selectivity for Ni/Sm and Co/Nd in table 3.

Author Response

The reviewer's comments were answered in a word document uploaded below.

Reviewer 2 Report

My comments are expressed in the attached document

Author Response

The reviewer's comments have been answered in a word document and uploaded below.

Reviewer 3 Report

The authors reported the adsorption of rare earth using surface-modified SiO2. The submission requires major revision before any consideration, including the following points:-

  1. The title should be revised to be informative and precise. The current title miss information such as adsorbent and main findings. The use for ‘Tailoring nano adsorbent surface’ and ‘recycling of Rare Earth-based magnets’ are confusing.
  2. Characterization techniques such as TEM nitrogen adsorption-desorption isotherm are required to characterize the materials’ porosity.
  3. The mechanism for adsorption should be further investigated
  4. A comparison with previously published adsorption SiO2 and other materials should be further discussed and summarized in a Table.
  5. References should be updated,, including these References Microporous and Mesoporous Materials 2019, 278, 175-184; Journal of Hazardous Materials, 2020, 386, 121632; Separation & Purification Reviews 2021, 50, 4, 417-444.
  6. The language should be revised, and typos should be corrected.

Minors

  1. All abbreviations such as ‘SEM-EDS, AFM, FTIR, TGA and tested for adsorption and separation of rare earth elements 10 (REE) (Nd3+ and Sm3+) and late transition metals (LTM) (Ni2+ and Co2+)’
  2. Correct typos such as ‘ 0,5-1,8mmol/g’,
  3. Add dash for ‘Rare Earth-based magnets’

Author Response

The reviewer's comments were answered and uploaded in the document below. 

Round 2

Reviewer 2 Report

This paper can be accepted in the present form. Congrats!!

Reviewer 3 Report

The authors addressed most of the comments and the revised version can be accepted.